# Seed Dormancy Class and Ecophysiological Features of *Veronicastrum sibiricum* (L.) Pennell (Scrophulariaceae) Native to the Korea Peninsula

**DOI:** 10.3390/plants11020160

**Published:** 2022-01-07

**Authors:** Gyeong Ho Jang, Jae Min Chung, Yong Ha Rhie, Seung Youn Lee

**Affiliations:** 1Department of Horticulture and Breeding, Graduate School of Andong National University, Andong 36729, Korea; jungle0158@gmail.com; 2Division of Plant Resources, Korea National Arboretum, Yangpyeng 12519, Korea; rhuso@korea.kr; 3Department of Horticulture and Forestry, Pai Chai University, Daejeon 35345, Korea; rhie@pcu.ac.kr; 4Division of Horticulture & Medicinal Plant, Andong National University, Andong 36729, Korea; 5Agricultural Science and Technology Research Institute, Andong National University, Andong 36729, Korea

**Keywords:** cold stratification, physiological dormancy, seed dormancy, seed germination, trait stasis, *Veronica*, *Veronicastrum*

## Abstract

*Veronicastrum sibiricum* is a perennial species distributed in Korea, Japan, Manchuria, China, and Siberia. This study aimed to determine the requirements for germination and dormancy break of *V. sibiricum* seeds and to classify the kind of seed dormancy. Additionally, its class of dormancy was compared with other *Veronicastrum* and *Veronica* species. *V. sibiricum* seeds were permeable to water and had a mature embryo during seed dispersal. In field conditions, germination was prevented by physiological dormancy, which was, however, relieved by March of the next year, allowing the start of germination when suitable environmental conditions occurred. In laboratory experiments, the seeds treated with 0, 2, 4, 8, and 12 weeks of cold stratification (4 °C) germinated to 0, 79, 75, 72, and 66%, respectively. After the GA_3_ treatment (2.887 mM), ≥90% of the seeds germinated during the four incubation weeks at 20/10 °C. Thus, 2.887 mM GA_3_ and at least two weeks at 4 °C were effective in breaking physiological dormancy and initiating germination. Therefore, the *V. sibiricum* seeds showed non-deep physiological dormancy (PD). Previous research, which determined seed dormancy classes, revealed that *Veronica* taxa have PD, morphological (MD), or morphophysiological seed dormancy (MPD). The differences in the seed dormancy classes in the *Veronicastrum*-*Veronica* clade suggested that seed dormancy traits had diverged. The results provide important data for the evolutionary ecological studies of seed dormancy and seed-based mass propagation of *V. sibiricum*.

## 1. Introduction

The most important role of seeds is to keep a species in existence. Accordingly, plants have evolved various strategies to ensure successful germination of seeds [1]. Germination is the first step in plant life history. The success of seed germination and seedling establishment can affect features for the propagation of plant species, which are of both economic and ecologic importance [2].

*Veronicastrum sibiricum*, which is endemic to Northeast Asia, is distributed in the central and northern parts of the Korean Peninsula, Far East Russia, Northeast China, Mongolia, and Northern Japan [3]. *V. sibiricum* is a perennial herb with a height of 50–90 cm, long, oval leaves, and light purple flowers in racemes blooming from July to August [3]. The species belongs to the Scrophulariaceae Juss., which includes approximately 220 genera and 4000 species, with 68 species in 25 genera distributed in South Korea. There are two forms of the same species of *Veronicastrum*, *V. sibiricum* and *V.*
*sibiricum* f. *albiflora* T. Yamaz in Korea [4]. The species of *Veronicastrum* and its closely related genus *Veronica* are widely distributed in the Northern Hemisphere, and in several regions in the Southern Hemisphere [5]. Both *Veronicastrum* and *Veronica* are morphologically closely related. *Veronicastrum* flowers have short calyx lobes and long corolla tube, whereas *Veronica* flowers have long calyx lobes and short corolla tube [6]. The phylogenetic analysis based on DNA sequences, seed microstructure [7], and pollen [8] indicate similarity between the two genera.

Traditionally, in South Korea, *V. sibiricum* roots have been used to treat neuralgia, arthritis, and inflammation and the young shoots are used as edible herbs [4]. Pharmacological studies have indicated the presence of compounds, such as isoferulic acid and 3,4-dimethoxy cinnamic acid, which have anti-inflammatory and analgesic properties [9], diterpene, which has antioxidant and anticancer properties [10,11], and iridoids for treating common cold, leucorrhea, cystitis, and liver [12]. 

Seed dormancy is a survival strategy, wherein germination is blocked under favorable environmental conditions [13,14]. Global data on seed dormancy and germination of 5250 species in major vegetation zones indicated that seeds of 69.6% of the species are dormant when freshly matured [15]. Strategies to break seed dormancy differ between species [16]. The seeds of many native plants in temperate regions have relatively small immature embryos [17]. If embryo elongation and germination stages in such immature embryos are achieved within 30 d under environmental conditions favorable for germination, the dormancy type is classified as morphological dormancy (MD) [13]. If there is an additional dormancy mechanism that inhibits the germination of immature embryos, the dormancy type is classified as morphophysiological dormancy (MPD). On the other hand, if a germination-inhibiting mechanism is added to seeds with mature embryos, the dormancy type is classified as physiological dormancy (PD) [13,15]. 

Choi [18] and Martinez-Ortega and Rico [19] determined that the genera *Veronica* and *Veronicastrum* are closely related based on the phylogeny results. Furthermore, the seeds of *Veronica parnkalliana*, a species native to South Australia, have immature embryos and show MPD, but 80–90% of the seeds germinated after gibberellic acid (GA) treatment [20]. Fifteen species of *Veronica* were classified as having PD, and *V. biloba* and *V. wormskjoldii* were found to have non-dormant seeds [13]. Song et al. [21] reported that eight species of *Veronica* native to the Korean Peninsula had MD or MPD. Some closely related species show the same dormancy type (trait stasis; [22]), whereas some exhibit significant differences in the type of depth of dormancy (trait divergence; [23]). This implies that during adaptation to different environments, dormancy patterns are either preserved or changed. Breaking seed dormancy and different germination traits are ecological characteristics and physiological control mechanisms of plants [22]. 

To determine the dormancy class, it is important to establish whether seeds contain embryos that are fully developed or underdeveloped at maturity and, therefore, if the embryo must elongate inside the seed before germination. Guerin et al. [20] and Song et al. [21] measured embryo growth in the seeds of *V. parnkalliana* and other *Veronica* species. However, many previous studies have not accurately measured growth (or the absence of growth) of the embryo in the seeds of species of *Veronica* and *Veronicastrum* [13]. Accurate classification of seed dormancy can provide a comprehensive understanding of the early stages of the plant life history and further facilitate efficient seed propagation. Therefore, this study aimed to investigate the seed ecology and physiological characteristics of *V. sibiricum* native to the Korean peninsula, to classify its seed dormancy, and develop technology for mass propagation based on the acquired ecophysiological information. Additional experiments to analyze the dormancy class of other species, which are closely related to *V. sibiricum*, were performed as a comparison.

## 2. Results

### 2.1. Internal and External Seed Morphology 

*V. sibiricum* seeds were light yellow and less than 1 mm in length (Table 1). The initial and prior to germination E:S ratios were 0.84 and 0.91, respectively (Figure 1), with no significant difference between the two E:S ratios (Figure 1).

### 2.2. Water Absorption Rate

Seed mass increased by more than 20% in 2 h compared to the initial weight, and increased by more than 30% after 24 h (Figure 2).

### 2.3. Phenology of Germination

This experiment was conducted to describe the phenology of germination from seeds kept under field conditions. The first germination event occurred on 26 March 2021 at daily average maximum and minimum temperatures of 13.1 °C and 4.5 °C, respectively. In the next two weeks, until 9 April 2021, 95% of the seeds germinated. Thus, in the natural environment, the seeds germinated about five months after reaching maturity (Figure 3). The time when seeds germinate is affected by the soil moisture content in natural conditions. In this experiment, there was little rain at the end of March, but there was a sufficient amount of rain at the beginning of April. For this reason, germination could rapidly increase in early April.

### 2.4. Effect of Light and Temperature on Germination

The germination test showed that only 10% of the seeds germinated at 25/15 °C under light conditions after 4 weeks of incubation, but they did not germinate at 4, 15/6, and 20/10 °C in light or dark (Figure 4).

### 2.5. Effect of Cold Stratification

A significant difference in the germination was found between the control group and the cold stratification treatment group. In the control group, germination was not observed, but the germination was more than 60% in all treatment groups after three weeks of culture (Figure 5). 

### 2.6. Effect of GA_3_ Treatment

The germination significantly increased with an increase in the GA_3_ concentration (Figure 6). In the first week of culture, the germination was 0, 6, 24, and 76% at 0, 0.029, 0.289, and 2.887 mM GA_3_, respectively; subsequently, in the fourth week of culture, the germination was less than 20% at 0 and 0.029 mM GA_3_, 45% at 0.289 mM GA_3_, and 90% or more at 2.887 mM GA_3_ (Figure 6). Thus, the germination was the fastest at 2.887 mM.

### 2.7. Effect of Light on Germination after Cold Stratification

The germination was more than 70% in the second week of culture under both light and dark conditions (Figure 7). Thus, the seeds could germinate even under dark conditions after cold stratification.

## 3. Discussion

Mature *V. sibiricum* seeds were sown in mid-October in phenology experiment. More than 90% of the *V. sibiricum* seeds germinated by the end of March in field conditions when the average maximum temperature for one week was 13.1 °C and the minimum temperature was 4.7 °C (Figure 3). *V. sibiricum* seeds did not germinate during winter and germinated in spring of the following year. In the low-temperature cold stratification treatment, germination occurred after two weeks (Figure 5). Therefore, in the natural conditions, the seeds may be in a state of ‘quiescence’ during winter [13], germination did not occur until late March of the following year due to unfavorable (low) temperatures for germination.

Soil seed bank is largely divided into transient type and persistent type. Transient type I seeds germinate in the summer or autumn of the year of seed detachment, whereas transient type II seeds spend the winter of the year of seed detachment in a dormant state, and then fully germinate during spring of the following year [24]. After the dormancy was broken, seeds germinated under dark conditions (Figure 7). Therefore, a short-term soil seed bank, which was classified as the transient type II, was established by suppressing seed germination in the same year.

Since mass of the *V. sibiricum* seeds increased by approximately 30% or more within 24 h compared with the initial mass due to water absorption (Figure 1), the seed coat was permeable to water [25] and the seeds do not have physical dormancy. 

The inner and outer shape of the embryos of *V. sibiricum* seeds observed in this study (Table 1 and Figure 8) were similar to those of the linear, axile embryos of the genus *Veronica* positioned in the center of the seed, which is classified as dwarf (seed size 0.3–2 mm) [17]. When seeds have an immature embryo that grows within 30 d under favorable germination conditions, they are considered to show MD [13]. As the difference in the E:S ratios between the initial point of seed detachment and immediately before germination were not significant (Figure 1), the embryo was considered to be in a mature state during detachment. Therefore, seeds of *V. sibiricum* do not have MD.

When *V. sibiricum* seeds were cultured for four weeks at various temperature conditions, only 10% germination was observed only at 25/15 °C (Figure 4), indicating that the seeds were dormant under appropriate environmental conditions [13]. If the seeds with mature embryos without PY or MD do not germinate within 30 d under favorable germination conditions, they are considered to exhibit PD [26]. Therefore, the dormancy type of *V. sibiricum* seeds was judged as PD. Based on the physiological germination inhibition mechanism, PD can be divided into three levels (non-deep, intermediate, and deep) [27]. Non-deep PD is broken by short-term warm or cold stratification or by GA_3_ treatment, intermediate PD is broken by cold stratification for at least two months and GA_3_ treatment, while deep PD is broken by cold stratification for 3–4 months but not by GA_3_ treatment [26]. *V. sibiricum* seeds did not germinate in the control that was not subjected to cold stratification, but germination was more than 60% in the group subjected to cold stratification for two weeks (Figure 5). Further, germination increased with the increase in GA_3_ concentration, and germination was more than 90% at 2.887 mM GA_3_ (Figure 6). Therefore, the dormancy was classified as non-deep PD, which was broken by a relatively short cold stratification period and by the GA_3_ treatment. We used the seeds stored for seven weeks at room temperatures in this study. Many seeds with PD show the effects of dry after-ripening that can release dormancy and promote germination [28]. Therefore, the depth of dormancy may decrease during storage. As a result of a preliminary experiment immediately after harvesting seeds in 2018, cold stratification for about 2 weeks was also effective in breaking PD of this species. This means that the relatively short cold stratification period is effective in breaking the PD. However, since we presented the results using seeds harvested in 2019 in this study, additional experiments are needed to know the exact depth of dormancy of seeds immediately after harvesting.

The subgenus *Pseudolysimachium* in the genus *Veronica* is assumed to have originated in Northeast Asia, and the differentiation of the genus may have occurred from Northeast Asia to Korea, Japan, China, and Europe [18]. Recently, several species of *Veronica* such as *V. dahurica*, *V. rotunda*, *V. kiusiana* var. *diamantiaca*, *V. pusanensis*, *V. rotunda* var. *subintegra*, *V. nakaiana*, *V. pyrethrina*, and *V. kiusiana* var. *glabrifolia* that are native to the Korean Peninsula have been placed in the *Pseudolysisimachion* reflecting the results of various molecular data [29]. In the Cenozoic Era, during which time the genus *Pseudolysisimachium* originated, Northeast Asia and North America were separated. Consequently, the subgenus was distributed locally only in the northern hemisphere, except in North America, due to the formation of a large mountain range in the lower regions of Northeast Asia [18]. Comparative analysis of chloroplast genomes in three species of the genera *Veronica* and the genus *Veronicastrum* revealed that the species had identical coding genes, tRNA and rRNA [30]. Therefore, the species of the two genera were considered phylogenetically closely related. We collected information on the class of seed dormancy of species of *Veronica* and *Veronicastrum* in previous research (Table 2). In previous studies, seed dormancy of *V. virginicum* and *V. parnkalliana* was classified as PD and MPD, respectively [20]. Song et al. [21] analyzed the ratio of embryo length to seed length (E:S ratio) in eight species of the genus *Veronica* (*V. dahurica*, *V. rotunda*, *V. kiusiana* var. *diamantiaca*, *V. pusanensis*, *V. rotunda* var. *subintegra*, *V. nakaiana*, *V. pyrethrina*, and *V. kiusiana* var. *glabrifolia*) native to the Korean Peninsula and found that the radicles in all eight species emerged after the E:S ratio increased from 18.8% to 58.0%. Among these species, seven had MD, and *V. kiusiana* var. *diamantiaca* had the characteristics of both MD and MPD at the population level [21]. Therefore, class of dormancy was differentiated at the species and genus level. Moreover, when the physiological characteristics of a species do not change during adaptation, the phenomenon is defined as “trait stasis.” Conversely, when physiological characteristics change during adaptation, the phenomenon is defined as “adaptation (or trait divergence)” [22]. Therefore, the findings of this study indicated that divergence occurred during dormancy in the closely related taxa because of the variation in embryo size and the different dormancy breaking requirements in species of the sister genera *Veronicastrum* and *Veronica*. The results of this study provide important preliminary data for the evolutionary ecological studies on seed dormancy and seed-based mass propagation in *V. sibiricum*.

## 4. Materials and Methods

### 4.1. Experimental Materials

The *V. sibiricum* seeds used in this study were harvested from the parent plant grown at the Andong National University affiliated farm (36°32′40″ N 128°48′03″ E) in Andong-si, Gyeongsangbuk-do on 16 October 2019. After harvesting, the seeds were dried in a laboratory at 21–27 °C for seven weeks before storing in a cold storage facility at 0 °C (DOI1815DOP; Winiamando, Gwangju, Korea) for further experimental use. Laboratory experiment was started at 2 weeks after storing the seeds at 0 °C. The seeds for phenology experiments to determine seed dormancy were harvested from the same area on 14 September 2020. 

### 4.2. Analysis of Internal and External Seed Morphology

To observe the internal and external morphology of seeds, the seeds were cut with a stainless steel razor blade (Dorco, Korea), and the resultant cross sections were photographed at 200–210× magnification under a USB microscope (AM3111 Dino-Lite premier; AnMo Electronics Co., Taiwan). Further, to determine whether the photographed seeds comprised immature embryos, embryos at the time of early detachment and just before germination were investigated, and the corresponding E:S length ratio was calculated.

### 4.3. Water Absorption

Water absorption was examined to identify the presence or absence of physical dormancy in the *V. sibiricum* seeds. Twenty seeds were placed in Petri dishes (90 × 15 mm; diameter × height) lined with two sheets of filter paper (ADVANTEC No. 1; Toyo Roshi Kaisha, Ltd., Tokyo, Japan) and added with distilled water (≒15 mL), and then cultured at 21–27 °C; four replicates were prepared per treatment. The seeds were not submerged in this experiment. The reason is that it was able to supply sufficient moisture without being submerged in our preliminary experiment. The distilled water was added as needed to keep the seeds moist. The initial weight of the seeds before water absorption and final weights after 2, 4, 8, 12, 24, and 48 h of water treatment were measured. The water absorption rate was calculated as follows:*W*_s_ (%) = [(*W*_h_ − *W*_i_)/*W*_i_] × 100
where *W*_s_ is relative weight ratio of seeds increased through water absorption, *W*_h_ is the weight per culture time, and *W*_i_ is weight of seeds before water absorption.

### 4.4. Phenology of Germination

This experiment was started after about 4 weeks of harvesting the seeds on 14 September 2020. The seeds were stored at the laboratory conditions before being used for the phenology experiment. The germination phenology experiment was conducted from 16 October 2020 to 16 April 2021 at the above-mentioned farm. There are several herbaceous plants growing around *V. sibiricum* in their native habitats. When the *V. sibiricum* seeds fall to the ground, direct sunlight was blocked by the plants. Therefore, a light-shielding film was installed where the seeds were planted to prevent direct sunlight exposure to the ground surface. Temperature data were collected using a sensor connected to a data logger (3683WD1; Spectrum Technologies, Inc., Aurora, IL, USA). The sensor was buried at a depth of 3 cm during the experiment, and the temperature was recorded every 30 min. 

To observe the seasonal germination response of the seeds, the germination was investigated weekly by exhuming the seeds in a phenology plot of the experimental farm. For the germination experiment, the seeds were wrapped in a plastic mesh filled with sand, and buried to a depth of 3 cm by filling the plastic pot with a horticultural substrate (Sunshine Mix #4; SunGro Horticulture, Agawam, MA, USA). This plastic pot was buried in the ground at a height similar to that of the top soil. Four replicates, each of 20 seeds per pot, were prepared. Once the emerged radicle reached a length of 1 mm, the seeds were regarded as germinated and were removed immediately.

### 4.5. Effect of Light and Temperature on Germination

The seeds to be cultured were sterilized in 500 mg·L^−^^1^ diluted solution of benomyl (a fungicide) for 4 h. Petri dishes (90 × 15 mm) were lined with two sheets of filter paper (ADVANTEC No. 1; Toyo Roshi Kaisha, Ltd., Tokyo, Japan), and 20 seeds were placed in each dish; four replicates were prepared per treatment. Distilled water (≒10 mL) was added to prevent the desiccation of the seeds. The seeds were cultured in a multi-room incubator (HB-101-4; HANBAEK-Scientific, Bucheon, Korea) at 25/15, 20/10, and 15/6 °C for 12/12 h of day/night and the photosynthetic photon flux density (PPFD) of 63.46 μmol·m^−^^2^·s^−1^. Culturing at 4 °C was conducted in a growth chamber (HB-603CM; HANBAEK-Scientific, Bucheon, Korea) under PPFD of 2.56 μmol·m^−2^·s^−1^. For incubation in dark conditions, petri dishes were wrapped in double aluminum foil to prevent the exposure of seeds to light. The seeds were examined each week and after four weeks, when the emerged radicle reached 1 mm in length, the seeds were considered germinated. Subsequently, the germinated seeds were counted and removed from the petri dish.

### 4.6. Effect of Cold Stratification

The effects of cold stratification were determined to understand whether the dormancy could be broken by cold stratification. Twenty sterilized seeds per replicate, four replicates per treatment, were placed in Petri dishes (90 × 15 mm) lined with two sheets of filter paper (ADVANTEC No. 1) and transferred to a growth chamber at 4 °C. Cold stratification at 4 °C was conducted in a growth chamber (HB-603CM; HANBAEK-Scientific, Bucheon, Korea) under PPFD of 2.56 μmol·m^−2^·s^−1^. After the cold stratification treatment for 0, 2, 4, 8, and 12 weeks, seeds were incubated in light at 20/10 °C.

### 4.7. Effect of GA_3_ Treatment

The effects of GA_3_ on dormancy break were analyzed by immersing the sterilized seeds in 0, 0.029, 0.289, and 2.887 mM GA_3_ solution for 24 h, after which they were rinsed with distilled water for 3 min and disinfected with benomyl wettable powder (500 mg·L^−1^) for 4 h. The seeds were sown in Petri dishes as described above and incubated at 20/10 °C. Distilled water was supplied to prevent desiccation.

### 4.8. Effect of Light on Germination after Cold Stratification 

The germination percentages based on varying incubation light conditions (light or dark) were determined. Twenty sterilized seeds per replicates, four replicates per treatment, were placed in Petri dishes and cold stratified at 4 °C for 2 weeks. The dishes of seeds in the dark treatment were wrapped with aluminum foil to protect the seeds from being exposed to any light. After the cold stratification treatment, both treatment groups were incubated at 20/10 °C for 4 weeks with 12/12 h of day/night and the photosynthetic photon flux density (PPFD) of 63.46 μmol·m^−2^·s^−1^.

### 4.9. Statistical Analysis

Analysis of variance was performed on the collected data using SAS 9.4 (SAS Institute Inc., Cary, NC, USA). Significant differences in embryo to seed length ratio (E:fvS ratio) at initial stage and seed coat split stage were determined by paired *t*-test at *p* < 0.05. To determine the statistical significance of the percentage germination between different treatments, Tukey’s honestly significant difference (HSD) test at *p* < 0.05 was used. The graphs were prepared using Sigma Plot 10.0 (Systat Software Inc., San Jose, CA, USA).

## 5. Conclusions

The seeds of *V. sibiricum* have fully developed embryos upon dispersal from their parent plants, and they exhibit non-deep PD. The seed dormancy was broken by more than 2 weeks of cold stratification at 4 °C. GA_3_ treatment substituted for cold stratification requirements: 90% of the seeds germinated after 4 weeks of incubation at 20/10 °C after a GA_3_ soaking treatment at 2.887 mM. There were differences in seed morphological and physiological traits between *Veronicastrum* and *Veronica* clade, indicating seed dormancy traits had diverged. The information obtained in this study could be used for commercial propagation of this plant and for the study of evolutionary mechanisms underlying seed dormancy in the *Veronicastrum* and *Veronica* clade.

## Figures and Tables

**Figure 1 plants-11-00160-f001:**
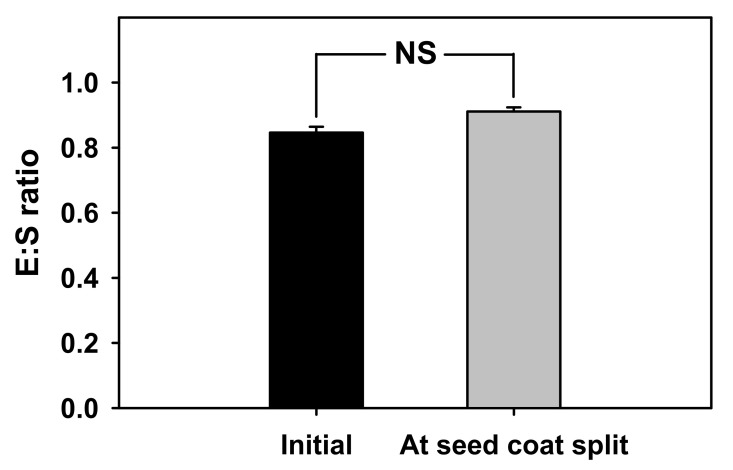
Embryo to seed length ratio (E:S ratio) at initial stage and seed coat split stage of *V. sibiricum*. Error bars are mean ± standard error (SE) of ten replicates. The NS represents no-significant differences as determined by paired *t*-test.

**Figure 2 plants-11-00160-f002:**
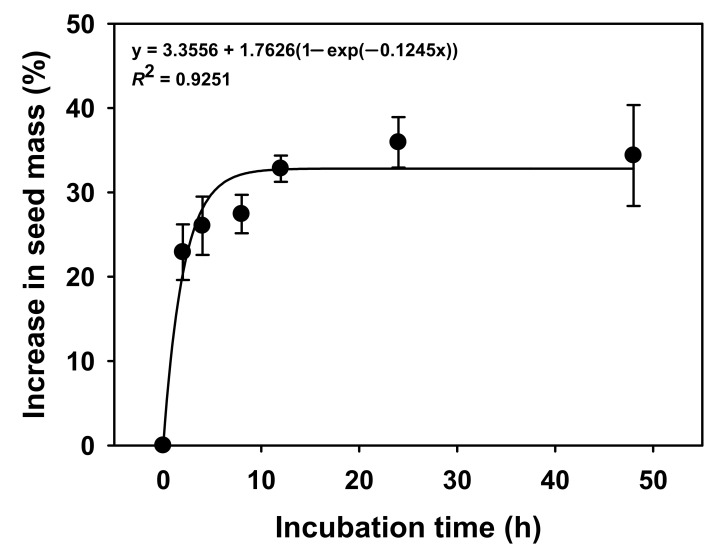
Water uptake by *V. sibiricum* seeds incubated at 21–26 °C on filter paper moistened with distilled water for 0–48 h. Error bars indicate mean ± SE of four replicates.

**Figure 3 plants-11-00160-f003:**
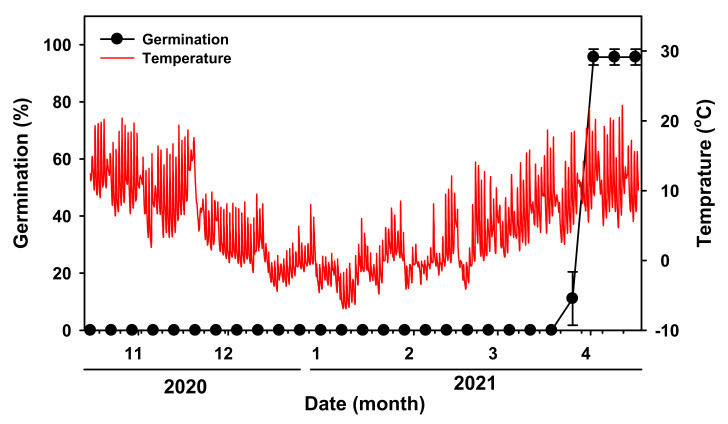
Phenology of germination of *V. sibiricum* seeds. The phenology experiment was conducted from 16 October 2020 to 16 April 2021. Vertical bars represent mean ± SE of four replicates.

**Figure 4 plants-11-00160-f004:**
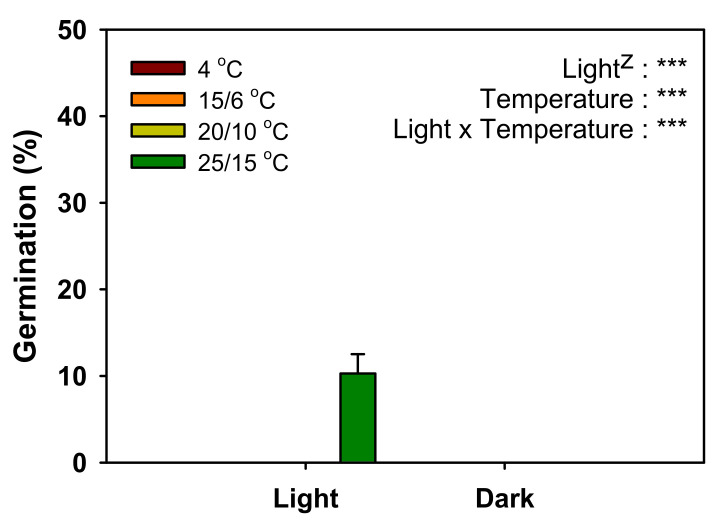
Germination of *V. sibiricum* seeds as affected by light conditions in response to four temperature regimes. Germination (%) was calculated at four weeks after incubation. Error bars indicate mean ± SE of four replicates. Results of a two-way analysis of variance applied to germination percentages of *V. sibiricum* seeds affected by light and temperatures (^***^ significant at *p* < 0.0001).

**Figure 5 plants-11-00160-f005:**
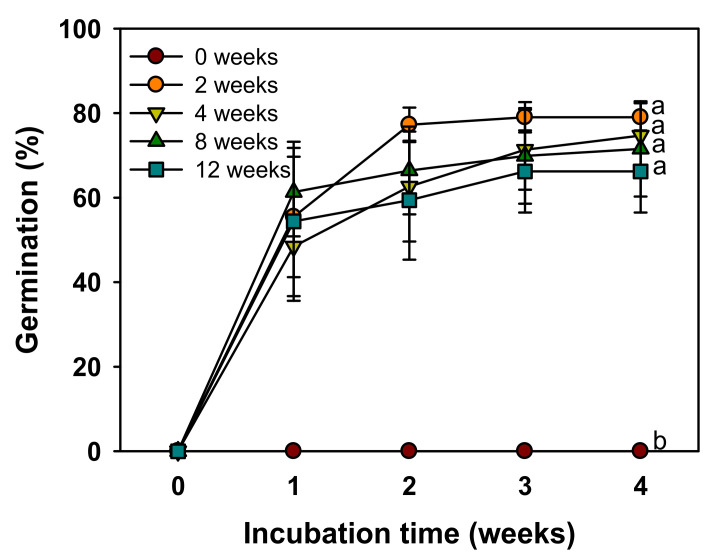
Germination of *V. sibiricum* seeds as affected by cold stratification periods (0, 2, 4, 8, or 12 weeks at 4 °C). Seeds were incubated at 20/10 °C after the stratification treatment. Error bars indicate mean ± SE of four replicates. The different letters represent statistically significant differences for germination percentages according to treatments in the fourth week as determined by the Tukey’s honestly significant difference (HSD) tests (*p* < 0.05).

**Figure 6 plants-11-00160-f006:**
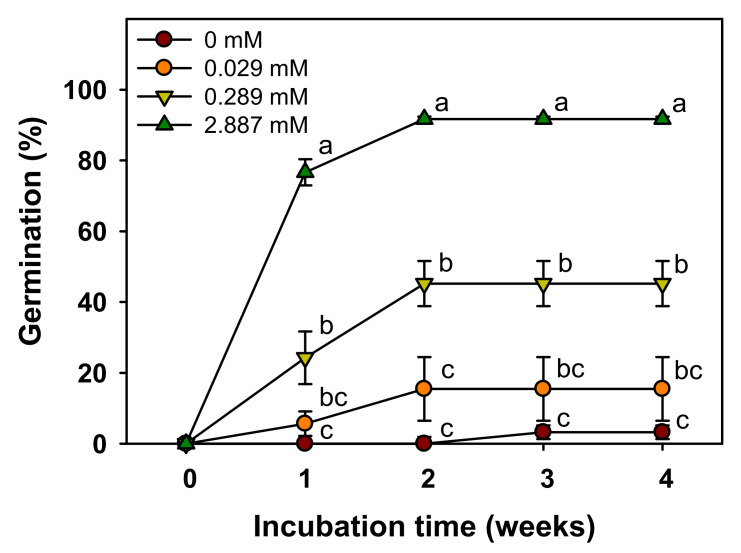
Effect of GA_3_ treatment on germination (%) in *V. sibiricum* seeds in light. Seeds were incubated at 20/10 °C (12/12 h). The seeds were soaked in 0, 0.029, 0.289, or 2.887 mM GA_3_ solution for 24 h. Error bars indicate mean ± SE of four replicates. The different letters represent statistically significant differences for germination percentages according to treatments in each week as determined by Tukey’s HSD tests (*p* < 0.05).

**Figure 7 plants-11-00160-f007:**
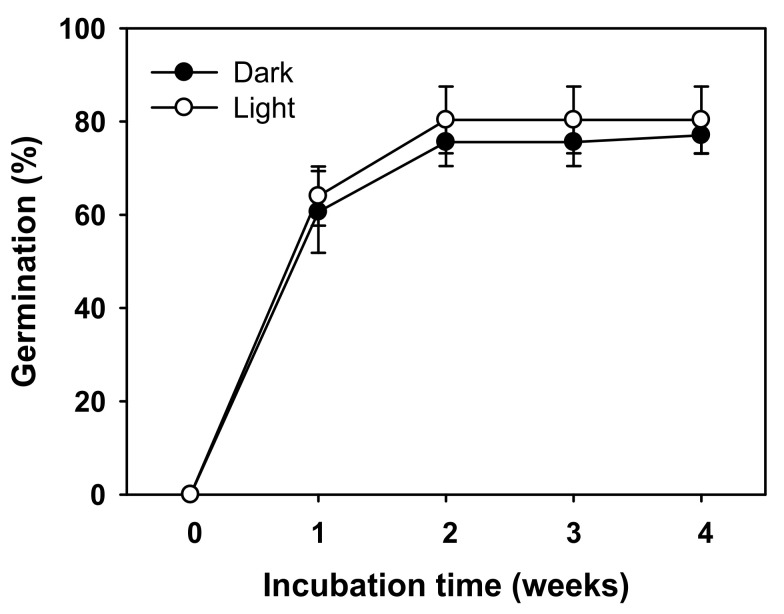
Germination of *V. sibiricum* seeds as affected by light after cold stratification at 4 °C for 2 weeks. Seeds were incubated at 20/10 °C (12/12 h) after cold stratification. Error bars represent mean ± SE of four replicates.

**Figure 8 plants-11-00160-f008:**
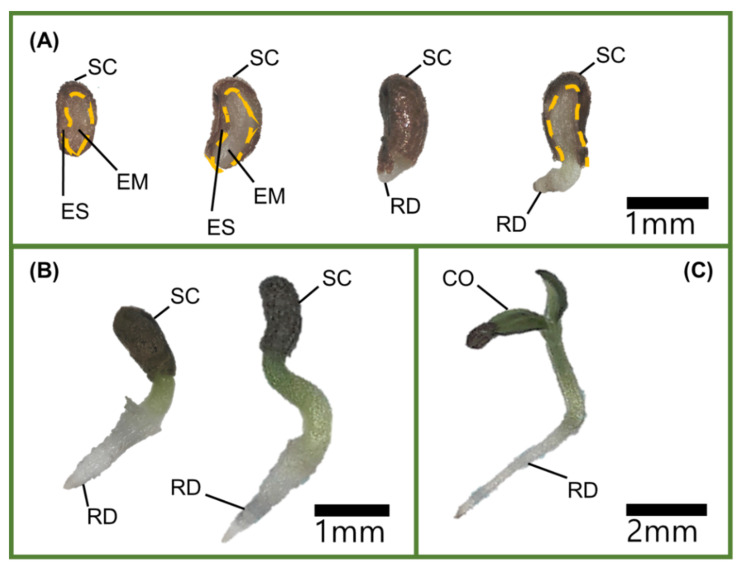
Germination and radicle emergence in seeds of *V. sibiricum*. (**A**) Seed cross section and germination (the top right seed) at early stages, (**B**) radicle emergence, and (**C**) shoot emergence. Scale bar is 1 mm. (SC = seed coat, EM = embryo, ES = endosperm, RD = radicle, and CO = cotyledon). The yellow line represents the boundary between the embryo and the endosperm.

**Table 1 plants-11-00160-t001:** Seed collection conditions and basic characteristics of *V. sibiricum*.

Scientific Name	Collection	Seed Length (mm)	Seed Width (mm)	100 Seeds Weight (mg)	Collection Date
*Veronicastrum sibiricum*	Collections of live specimens from genetic resources plot, Andong National University, Korea	0.75 ± 0.010 ^z^	0.05 ± 0.010 ^z^	7.74 ± 0.200 ^y^	16 October 2019
		-^x^	-	-	14 September 2020

^z^ mean ± SE (*n* = 10). ^y^ mean ± SE (*n* = 3). ^x^ not measured.

**Table 2 plants-11-00160-t002:** Seed dormancy classes in *Veronica* and *Veronicastrum* taxa.

Taxa	Native to ^z^	Dormancy Classes	References for Dormancy Classes
*Veronicastrum sibiricum*	Siberia to Japan and N. China	PD ^y^	[13]
*Veronicastrum virginicum*	Central and E. Canada to Central and E. USA	PD	
*Veronica americana*	Central and E. Canada to Central and E. USA	PD	
*Veronica arvensis*	Macaronesia, NW. Africa, Europe to SW. Siberia and W. Himalaya	PD	
*Veronica hederifolia*	Macaronesia, Europe to Medit. and Central Asia	PD	
*Veronica officinalis*	Macaronesia, Europe to W. Siberia and N. Iran.	PD	
*Veronica peregrina*	North and South America	PD	
*Veronica persica*	Caucasus to N. Iran	PD	
*Veronica cana*	Himalaya to China (NW. Yunnan) and N. Myanmar	PD	
*Veronica wormskjoldii*	E. USA to Greenland	ND	
*Veronica parnkalliana*	SE. South Australia	MPD	[20]
*Veronica dahurica*	Korea	MD	[21]
*Veronica rotunda*	Korea	MD	
*Veronica pusanensis*	Korea	MD	
*Veronica rotunda* var. *subintegra*	Korea	MD	
*Veronica nakaiana*	Korea	MD	
*Veronica pyrethrina*	Korea	MD	
*Veronica kiusiana* var. *glabrifolia*	Korea	MD	
*Veronica kiusiana* var. *diamantiaca*	Korea	MD + MPD	
*Veronicastrum sibiricum*	Siberia to Japan and N. China	Non-deep PD	Present study

^z^ Native ranges of each taxon are based on the data of ‘Korea Biodiversity Information System (KBIS), Korea National Arboretum’ or ‘Plants of the World Online, Kew’. ^y^ PD: physiological dormancy, ND: non dormancy, MPD: morphophysiological dormancy, MD: morphological dormancy.

## Data Availability

Not applicable.

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
