# Peer review of "Seed Dormancy Class and Ecophysiological Features of Veronicastrum sibiricum (L.) Pennell (Scrophulariaceae) Native to the Korea Peninsula"

_plants, 2022, doi:10.3390/plants11020160_

Round 1

Reviewer 1 Report

The paper concerns the important aspect of wild plants survival strategy -  seed dormancy. Veronicastrum sibiricum has a drfinite value for pharmacology and even as a food plant. Therefore, it is important to understand the conditions for its controlled propagation. The authors achieved their goal by studying the nature (type) of seed dormancy and the conditions for its breakage.

Reviewer 2 Report

Reading this manuscript went quite smoothly until I came across this sentence, which made me gasp: “After harvesting, the seeds were dried … at 21–27 °C for seven weeks before storing them in a cold storage facility at 0 °C … for further experimental use” (lines 260-262). Seeds that have nondeep physiological dormancy become nondormant during dry storage at room temperatures, a phenomenon known as dry after-ripening (Baskin and Baskin 2020. Breaking Seed Dormancy during Dry Storage: A Useful Tool or Major Problem for Successful Restoration via Direct Seeding? Plants 9:636). In particular, those authors (Baskin and Baskin 2020) write that “afterripening at room temperatures will be relatively rapid, that is, one to three months”, with specific reference to their previous study on Veronica (Baskin and Baskin 1983. Germination ecology of Veronica arvensis. J. Ecol. 71:57–68). As Veronica and Veronicastrum are closely related genera (lines 76-77), there is no doubt that drying seeds at 21–27 °C for seven weeks prior to study their dormancy features was a very bad idea, also given that, with ventilation, one to two days are well enough to dry these seeds. Besides, below zero temperatures are typically recommended to preserve dormancy in dried orthodox seeds.

As it appears very probable that the seeds used in this study had some dormancy relieved before they were studied for the “Seed Dormancy Type” (from the title), it is clear that part of the present results are flawed. Yet, the main conclusions of this study still hold. In fact, as noted above, dry after-ripening occurs if the seeds have nondeep physiological dormancy, which, indeed, is the main conclusion of this manuscript. So, what is unreliable is just the absolute values of germination. The authors must, therefore, carefully revise the manuscript highlighting that their data do not reflect the original dormancy level of freshly harvested seeds (what a pity!), and they must provide, in the manuscript, a reasonable justification for this. In particular, the authors must explain, in the manuscript, why they neglected the role of dry after-ripening in the eco-physiological study of this seeds; a fact that, in my opinion, was an awful – though not fatal, in this case – mistake.

Minor comments are detailed below.

Lines 39-40, “Germination is the first-point in plant life history, which decides when and where plants will grow”: germination cannot decide where plants grow.

Lines 47-48: they are two forms of the same species.

Please, note that even when a genus name appears on its own it must be italicized. So, “Veronicastrum” and “Veronica” should always be in italics. Please, amend throughout the manuscript.

What do the authors think of the fact that Veronicastrum, like Veronica, has been recently moved to the Plantaginaceae (e.g., ref. 5, 7, 27)?

Lines 50-52: I do not understand what the difference between the corolla of Veronicastrum and Veronica is, and I do not think the description of the corolla of Veronica is right.

Line 53: I do not understand what the authors mean by “DNA sequences of the microstructures of the seeds”. Please, re-phrase this sentence.

Lines 57-58: what is “dimethoxy”?

Lines 59-62: the mentioned species is not the same as V. sibericum, and, as far as I know, very many plants with entomophily are “a significant nectar source for pollinators” because the insects visit them just to collect nectar.

Lines 63-65, “Seed dormancy is a survival strategy, wherein germination is obstructed under un-favorable environmental conditions; an appropriate germination time is detected when conditions for growth improve”: first, change “obstructed” to “blocked” or alike. Second, “dormancy” occurs when living seeds do not germinate under conditions that are otherwise favourable for germination. The state in which seeds do not germinate due to unfavourable environmental conditions is called “quiescence”. So, please, do not confound these two concepts. Third, “an appropriate germination time is detected” is not understandable; please, re-phrase.

Regarding the second point above, see ref. 13. More specifically, dormancy has been proposed to be the eco-developmental quiescence of a meristematic or embryonic organ, whereby growth fails to respond to favourable conditions until sufficient entrainment by environmental cues occurs (Considine and Considine, 2016, On the language and physiology of dormancy and quiescence in plants. J. Exp. Bot. 67: 3189–3203).

Line 78, “seeds … show MPD and have immature embryos”: that they have immature embryos is already implied by their having MPD, specifically, by the “M” (morphological). So, it would better to say: “seeds … have immature embryos and show MPD”.

Subsection “Phenology of germination” (line 126 and following): as the set-up of this experiment is described in the Methods section, at the end of the paper, the experiment should be briefly introduced, since the reader does not know anything about it, at this point. This holds true for all the experiments, but the previous ones are easier to understand, even without any additional description.

Lines 129-130, “the seeds germinated after maturity after five months in the natural environment”: first, this sentence is not entirely clear; please, improve the wording. Second, seed germination also depends on water availability. I suppose the soil always had enough water during the Winter, but, please, clarify this aspect in the manuscript.

Line 178, “V. sibiricum seeds remained dormant in the ground during winter”: or, they were initially dormant, but become quiescent after a few weeks of cold stratification in the soil. In fact, “endo-dormancy broke during early winter, germination occurred at the end of March of the following year due to unfavorable temperatures for germination” (lines 181-182). However, these are authors’ hypotheses, since they did not exhumate the seeds to ascertain when they were dormant and when they were just quiescent due to too low temperatures for germination.

Line 275, seeds were placed in “a 90 × 15 mm petri dish; subsequently, distilled water was added”: first, “Petri” ought to have the first letter capitalized, as it is the name of the guy who devised this kind of container. Second, “90 × 15 mm” seems to suggest that these were rectangular dishes. Was it so? Third, please, clarify, in the manuscript, how much water was added, and if it was subsequently replenished. All this applies to the corresponding description of the other laboratory experiments.

As for the statistical analysis, “Analysis of variance was performed on the collected data collected using SAS” (line 333). Germination binomial data violate ANOVA assumptions, and, therefore, a suitable transformation is often recommended. Besides, the analysis of germination time-courses requires a more specific statistical approach, like the repeated-effect ANOVA, or even better, generalized linear mixed models, which are suitable to deal both with binomial data as well as with repeated observations through time, and are available in SAS (see, for example, Gianinetti 2020. Basic features of the analysis of germination data with generalized linear mixed models. Data 5:6).

Reviewer 3 Report

Authors investigated the requirements for the germination and dormancy break in seeds of Veronicastrum sibiricum the endemic species to Northeast Asia. They revealed that non-deep physiological dormancy (PD) is characteristic to this species. This species is interesting to study because traditionally, in South Korea, V. sibiricum roots have been used to treat neuralgia, arthritis, and inflammation and the young shoots are used as edible herbs. Additionally Authors compare the dormancy patterns in the genus genera Veronica  and Veronicastrum.

MAJOR POINTS:

  1. Introduction lines 75-107, the two paragraphs are the review of literature and suits to discussion section. Please consider shorting this part to the most informations that will introduce readers to this topic and move more detailed informations to the Discussion section.
  2. Figures.:
    Figure 1. Add in the caption that differences were not significant or add the same letters to point out that statistic analyses revealed that.

Figure 4. The additional name of x axis is not necessary, “dark” and “light” are sufficient.

Figure 6. Add in legend that the concentration relates to GA. Statistics is missing for the seeds incubated for 3weeks.

Figure 5 Add in caption what dark/light conditions were used to perform germination when testing different length time of  cold treatment? How does data from figure 5 (2 weeks treatment) differs from these given in Figure 7 in the dark?

Figure 8. What do the yellow lines mean? Please explain.

  1. Germination test. Authors wrote in line 323 that “20 seeds were placed in the petri dish for four replicates”. ISTA rules recommend 4 replication of 100 seeds. Some studies use 4 replications of 50 seeds but only when seeds are very big (larger than 1-2 cm). Such sample is too small because each seed is the equivalent of 5% of changes. Please provide data on larger seed samples.
  2. Conclusion is lacking. Please write it providing a protocol what is the simplest/fastest/most efficient procedure for successful germination in this species.

MINOR POINTS

Line 40, correct “history , which”

Lines 129/130, check the font, it seems to be different

Line 281 correct “wi”

Reviewer 4 Report

Line 2. Change “type” to “class”.  You are studying a class of dormancy and not a type of physiological dormancy

Line 19. Change to “ “classify the kind of seed dormancy.” “Additionally, its class of dormancy was compared…”

Line 29. Change “types” to “classes”

Line 37. Seeds cannot think or plan.  Change wording to “Sexual production by seeds is an important aspects in the maintenance of the vigor of a plant species. “

Line 38. Change to “strategies that successfully implement a genetic…”

Line 40. Change “decides” to “determines”

Line 45. Delete “family”    not needed

Lines 47 and 50. Delete “the genus” and put “Veronicastrum” in italics

Line 48. Delete “genera” and put “Veronicastrum” and “Veronica” in italics

Line 50.  Also  delete “the genus”  and “corolla”

Line 51. Delete “the genus” and put “Veronica” in italics

Line 54. Change “in” to “between”

Line 61. Delete “the genus” and put “Veronicastrum” in italics  

Line 61. Change “gas” to “have” and change “is also” to “they also are”

Line 63. Change “obstructed” to “prohibited”

Line 66. Change wording to “indicated that seeds fo 69.6% of the species are dormant when freshly matured.”

Line 67. Change “in each” to “between”

Line 73. Change to “if a germinating-inhibiting mechanism…”

Lines 76, 77, Put names in italics

Line 80. Delete “the genus” and put “Veronica” in italics

Line 84. Delete “the genus” and put “Veronica” in italics

Line 84. Change “the results showed” to “found”

Line 85. Change “ratios” to “ratio”

Line 86. Change “showed” to “had”

Line 88. Change wording to “related species had the same class of dormancy, whereas others exhibited …”

Line 89. Change to “differences in the class of dormancy or depths.”

Line 94. Change to “the classification of seed dormancy, checking…”

Line 95. Change to “ the seeds elongate prior to radicle emergence is important.”

Line 96.  Change to “measured embryo growth in the seeds of….”

Line 97.  Change to “measured growth (or not) of the embryo …”

Lines 96-97 and 98.  Put names in italics

Line 98. Change to “seeds of species of Veronica and”

Line 101. Change to “classification of seed dormancy can provide a comprehensive…”

Line 104. Change to “classify its seed dormancy”

Line 105. Delete “types” and change wording to “ develop technology for mass…”

Line 106. Change to “addition, seed dormancy class of v.”

Line 107. Change “were” to “was”

Line 113. Change “for” to “of”

Line 120.  Delete “rate”  Rate means the speed of germination, but you are not talking about speed.  You are talking about % of germinate – two very different things.

Line 129-130. Delete “and the seeds germinated after maturity after five months in the natural environment” 

Line 135. Delete “according to light and temperature conditions”

Line 127. Insert “they” after “but”  

Line 127. Insert “in light or dark” after “201/0 C” and delete the rest of the sentence.  Keep “(Fig.4).”

Lines 138-139. Delete the sentence that begins with “Moveover”

Line 151.  Insert “in light” after “seeds”     Is this correct?

Line 157. Change “rates were” to “was”

Line 163. Insert “in light” after “seeds”     Is this correct?

Line 168. Delete “rate”

Lines l64 and 172. Delete “conditions”

Line 181. Change to “germination did not occurred until the end of…”

Line 182. Insert “(low)” after “unfavorable”

Lines 186-188. Delete “In this study germination did not occur in the temperature experiment under dark conditions (Fig. 4), and”.  Begin a new sentence with “After the dormancy…”

Line 191. Change to Since mass of the V. sibiricum seeds increased…”

Line 192. Change “weight” to “mass”

Lines 192-193. Change to  “(Fig. 1), the seed coast was permeable to water [24} and the seeds do not have physical dormancy.”

Line 196. Put name in italics

Line 202. Change to “therefore, seeds of V. sibiricum do not have MD.”

Line 209. Change to “conditions only 10% germination was…”

Line 215. Change to “PD is broken by short-term….”   “or by GA3…”

                “intermediate PD is broken by cold”

               “deep PD is broken by  cold…”

Line 219. Delete “group”

Lines 220, 221, 222. Delete “rate”

Line 223. Delete “type”

Line 225. Put names in italics

Line 227. Change to “several species of Veronica…..”

Line 230. Change to “have been placed in “  Put name in italics l

Line 231. Change “which the” to “which time”

Line 232. Put name in italics

Line 236. Put names in italics

Line 238. Change to “information on class of seed dormancy of species of Veronica and Veronicastrum in previous research (Table 2).”   “Seed dormancy of V. virginicum….”

Line 242. Put name in italics. And delete “one”

 Line 243. Change Showed” to “had”   Change wording to “therefore, class of dormancy was …”

Line below 253 (in the table).   Change “types” to “classes” in both places

Line 261.  I see that you stored seeds at room temperatures for 7 weeks before you put them in cold storage.  Seven weeks is long enough for the seeds to come out of dormancy via after-ripening.  Need to add some discussion about after-ripening to the Discussion.  Not using fresh seeds is a serious problem in trying to determine the class, level and type of dormancy seeds have.   How long were seeds held in cold storage before they were used in experiments?  Based on the methods presented in the manuscript, it would be very hard for anyone to repeat your work.  Need to give enough details to make it easy for the work to be repeated.

Lines 275, 307, 310, 323, 327.  Change to “Petri”  This is a man’s name; invented the Petri dish.

Line 282.  The methods for phenology of germination are not clear.  I cannot follow what you did.  Where seeds plants on the same day they were collected?  How could you monitor germination is the seeds were covered by 3 cm of soi?   Did you have to dig up the seeds each time you monitored germination?  why not sow the seeds on the soil surface?  Need to provide more information.

Line 297.  Need to tell reader how old the seeds were when you started this experiment.

Line 315. Were the seeds in light or dark?  Need to add this information to the Methods.

Line 316-317.  Change to “12 weeks, seeds were incubated in light at 20/10 C”  Is this what you did?

Line 326. Change “rates” to “percentages”

Line 327. Change to “placed in Petri dishes and cold stratified at 4 C for 2 weeks.”  “The dishes of seeds in the dark treatment were wrapped…”

Line 329 and 330. Change to “ seeds from being exposed to any light. After the  cold…… incubated at 20/20 C for 4 weeks.”

Reviewer 5 Report

This is potentaially inetersting point of view, but text require significant corrections. Please, made some diagramm to describe your experiments, sometimes it is not so easy to follow.

Please, read each sentence and make it more clear (examples lines 183, 187).

Some details:

Line 22:  what due to dormancy?  Please, re-formulate.

Line 27: From the previous researches ??

Line 28: PD? Please, provide full name.

Line 52: „The phylogenetic analysis using DNA sequences of the microstructures of seeds [7] and pollen [8]“ ?  what do you mean as „DNA sequences of the microstructures“?

In the introduction it will be importnat to mention also this paper: https://nph.onlinelibrary.wiley.com/doi/full/10.1111/j.1469-8137.2006.01787.x

And dormancy – germination transition as chormatin remodelling.

Line 85: „the radicles in all eight species emerged after the E:S ratios increased“ – I am not sure after, I think it is better to write correlated, not after.

Figure 1: seeds coat split may means that seeds alraedy became active (active chormatin and gene expression). It will be nice in future to test can such seeds germinate after completely drying.

Line 127: please, induicate full time-tables of your experiment: „March 26, 2021“ is OK, but when your soaked seeds?

Figure 4 is not informative. One column only!

It is better to use table instead.  What is LihgtZ?

Figure 6: 1 g7L GA3 is not a hormone-related concentration. Have you measure pH oft he GA solution?  

Figure 8: plesae, clarify what do you mean as gerimation (how did you count). On figure 8 nice pictures were shwon, but which seeds stage you count as germination?

Actually, once chromatin became active and seeds resources starting to used, seeds can be sonsider as germinated. Is it the case in your model?  

Line 183: „Soil seed bank is largely divided into transient type and persistent type“ – It looks like seeds in the bank, but not bank divided.

Line 187: „germination did not occur in the temperature experiment under dark conditions“ – not clear what do you mean, please, re-formulate.

„Effect of incubation light condition and culture temperature“ – plesae, re-formulate: you do not need to write „incubation light conditions“, light conditions is enough.

Line 319: 1 g/l GA3 is quite out of hormone range (almost 3 mM, not µM). How can you exclude effect of simple pH (3 mM solution must be acid)?

Round 2

Reviewer 2 Report

My main concern about the present study was, and still is, that the seeds had some dormancy relieved before they were studied. As I required, the authors highlighted that their data do not reflect the original dormancy level of freshly harvested seeds, and, thus, “additional experiments are needed to know the exact depth of dormancy with seeds immediately after harvesting” (lines 218-219). I also required, however, that they provide, in the manuscript, a reasonable justification for having after-ripened these seeds without apparent needing. The fact that the change in the depth of physiological dormancy was slight as “the intermediate PD type requires low-temperature stratification for 2-3 months to break the PD” (from the authors’ response) is, at present, not supported, because they conclude that these seeds “exhibit non-deep PD” (line 345), not intermediate PD. So, please, provide a reasonable explanation to the readers. Please, also note that, though dry after-ripening could be important in the eco-physiological characterization of this species, it is just mentioned once in the manuscript (on line 217).

In this regard, I suggest changing, in the title, “Implications” to “Features”; in fact, it is not clear what “Implications of Veronicastrum sibiricum” means.

Please, note that incubating seeds, particularly this small, in 90-mm-diameter Petri dishes with 15 mL of water (lines 276 and 304), seems really too much water: 3-5 mL are commonly used for these dishes. Of course, it depends on the kind of filter paper, but please, clarify, in the manuscript, whether the seeds were submerged or not. Submergence is not an optimal condition for germination and would negatively affect all the results. So, the authors must clearly explain, in the manuscript, why they performed the germination tests with this amount of water; this is also necessary for the future replicability of their findings.

On lines 174-175, it is stated that “germination did not occurred until the end of March of the following year due to unfavorable (low) temperatures for germination”: this is in contrast with what said on lines 123-126. Please, be consistent.

On lines 21-22 and 169-170, the authors claim that “seeds germinated by the end of March in their natural conditions”: as “A light-shielding film was installed where the seeds were planted to prevent direct sunlight exposure to the ground surface” (lines 287-288), I do not think these can be defined “natural conditions”. Please, clarify this aspect, in the manuscript.

In addition, what does “naturally dispersed” (lines 21-22) means, in the present context?

As remarked by another reviewer, please, note that the data shown throughout the manuscript are germination percentages, not germination rates. The germination rate is the reciprocal of the time to germination (Bewley and Black, 1994, Seeds: Physiology of Development and Germination, 2nd ed., Plenum Press, New York; Ranal and de Santana, 2006, Brazilian J. Bot. 29:1-11). Please, amend throughout the manuscript (e.g., lines 126, 142, 150, 161).

As for the statistical analysis, “Analysis of variance was performed on the collected data using SAS” (line 337). As already said, germination binomial data violate ANOVA assumptions, and, therefore, a suitable transformation is often recommended. So, please, explain why you did not deem necessary to apply any data transformation, or apply it. Besides, as said as well, the analysis of germination time-courses requires a more specific statistical approach, like the repeated-effect ANOVA, specifically in Figure 6, because the subsequent observations are not independent. This is particularly relevant in the present study, since “To observe the seasonal germination response of the seeds, the germination was investigated weekly by exhuming the seeds”, it seems, therefore, that the seeds were regularly exhumed and re-buried (please, clarify, in the manuscript, if this was indeed so), which can affect the overall trend of the germination curve. Unless, of course, you use the same statistical trick of Figure 5, which I do not like, but is not entirely wrong.

As it happens, several language issues have been introduced during the revision; so, language editing and proofreading is needing. Some suggestions are given below.

Line 19: change “and class the kind of seed dormancy” to “and to classify the kind of seed dormancy”.

Lines 22-23, “and germination started in March of the next year due to physiological dormancy”: as we agree that dormancy was probably removed earlier than March, I’d say “and germination was prevented by physiological dormancy, which was, however, relieved by March of the next year, allowing the start of germination when suitable environmental conditions occurred”.

Lines 37-38, I’m sorry to say that “Sexual production by seeds is an important aspects in the maintenance of the vigor of a plant species” is not a good sentence: “Sexual production” does not occur by means of the seeds, rather, seeds, like babies, typically are the product of sexual reproduction; “vigor” has a specific meaning in the seed sciences, which does not fit here; and the overall meaning of this phrase is not clear.

Analogously, “plants have developed various survival strategies that successfully implement a genetic transfer system and to germinate seeds safely” (lines 38-39) is not clear: what exactly is meant with “a genetic transfer system”? What is a survival strategy “to germinate seeds safely”?

Lines 39-41, “Germination … affect the postgerminative growth of the seedlings” is a triviality: if a seed does not germinate there is no post-germinative growth at all. Please, re-write this whole paragraph (lines 37-41).

Lines 47-48: if there is “disagreement among taxonomists”, please, decide whether to mention them as two forms of the same species or two different species. As, in the binomial system of naming species, each name has two parts, the genus and the species, if the latter part is the same across two names, they correspond, by definition, to the same species. So, if you mention them as two forms of the same species (sibiricum), please, call them “two forms of the same species”. Easy, I’d suppose.

Line 51: delete “of”.

Line 54: delete “the” before “similarity”.

Line 61: change “prohibited” to “blocked”, or “prevented”, or “restrained”, or “restricted”, or “repressed”, or “halted”.

Lines 70-71, change “Further, if a germinating-inhibiting mechanism is added to seeds with mature embryos …” to “On the other hand, if a germination-inhibiting mechanism is added to seeds with mature embryos …”.

Line 85, change “During the classification of dormancy, it is important aspects whether …” to “To determine the dormancy class, it is important to establish whether …”.

Line 89, change “growth (or not)” to “growth (or the absence of growth)”.

Lines 90-92, “V. sibiricum may have been classified as showing MD or MPD with immature embryos because many closely related species have similar dormancy patterns”: but this witless thing, fortunately, did not happen; therefore, delete this sentence.

Lines 97-98, “In addition, seed dormancy class of V. sibiricum and other closely related species was compared and analyzed” seems to say that you did additional experiments to analyse the dormancy class of other species, closely related to V. sibiricum, as a comparison. Please, re-word this sentence.

Line 106 (footnote of Tab. 1), “(n = 3)”: this means three 100-seed samples, rather than three seeds, I suppose. Please, clarify.

Line 118: please, make clear that this experiment was conducted in the field.

Lines 125-126: change “For this reason, it is interpreted that the germination rate increased rapidly in early April” to “For this reason, germination could rapidly increase in early April”.

Line 132: did germination only occur “in the fourth week” of incubation?

Line 168, “Mature V. sibiricum seeds were sown in mid-October (Table 1). Consequently, …”: Table 1 shows the date of collection, not of sowing. Besides, remove “Consequently”, as there is no direct consequence between sowing date and what is said thereafter. Furthermore, the experiment of Fig. 3 (Phenology of germination) is referred to here, but, in the Methods, it is stated that “This experiment was started after about 4 weeks of harvesting the seeds on September 14, 2020” (lines 285-286). Something is wrong, here.

Line 173: Fig. 5 should be referred to, in place of Fig. 3.

Lines 212 and 213: remove “the” before “germination”.

Line 218: change “vary” to “decrease”.

Lines 219: change “with” to “of”.

Line 233: “on the class”.

Line 297: “replicates”.

Line 298: “regarded”.

Line 318: “transferred”.

Line 348: “There were”.

Reviewer 3 Report

The manuscript was significantly improved and is ready to be published.

Reviewer 5 Report

Thank you very much for detailed corrections.

Some questions still remained.

Lines 149-159: what was pre-treatments for GA3 experiments? Please, explain.

Figures 5,6 and 7: it will be informative to shown in insert kinetics of germination for each day during first week.

It also will be useful for the future experiments shown kinetics of chirmatin remodeling directly and indirectly. Directly by investigation of chromatin status at each day before radicle emergence and indirect by completely drying seeds after 0-48 h incubation in water and new germination test. While it is commonly accepted among practical scientists that seeds germination is a radicle emergence, the process of seeds germination is dormancy broken what related with chromatin relaxation (aftzer 12-24 h after soaking). Thereafter seeds can do go back to deep dormancy and, if drying comopletely, can not germinate anymore.

Line 76: 1 g/l GA3 can’t be considered as Gibberellic acid. Gibberlleic acid is a hormone what shoiukld work in µmol concentration. It looks like it can be non-specific effect (may be as acid) and control need to be used for this case.     

Line 318: “transferred”
